# Research on Similarity Simulation Experiment of Mine Pressure Appearance in Surface Gully Working Face Based on BOTDA

**DOI:** 10.3390/s23229063

**Published:** 2023-11-09

**Authors:** Dingding Zhang, Zhiming Huang, Zhe Ma, Jianfeng Yang, Jing Chai

**Affiliations:** 1College of Energy Engineering, Xi’an University of Science and Technology, Xi’an 710054, China; huangzm@stu.xust.edu.cn (Z.H.); mz13088958726@stu.xust.edu.cn (Z.M.); yangjianfeng@xust.edu.cn (J.Y.); chaij@xust.edu.cn (J.C.); 2Key Laboratory of Western Mine Exploitation and Hazard Prevention, Ministry of Education, Xi’an 710054, China

**Keywords:** BOTDA, optic fiber frequency shift variation, discrimination of strong ground pressure, similar physical model experiment, precursor characterization of instability

## Abstract

In order to study the mountain deflection characteristics and the pressure law of the working face after the mining of a shallow coal seam under the valley terrain, a geometric size of 5.0 × 0.2 × 1.33 m is used in the physical similarity model. Brillouin optical time domain analysis (BOTDA) technology is applied to a similar physical model experiment to monitor the internal strain of the overlying rock. In this paper, the strain law of the horizontal optical fiber at different stages of the instability of the mountain structure is analyzed. Combined with the measurement of the strain field on the surface of the model via digital image correlation (DIC) technology, the optical fiber strain characteristics of the precursor of mountain instability are given. The optical fiber characterization method of working face pressure is proposed, and the working face pressures at different mining stages in gully terrain are characterized. Finally, the relationship between the deflection instability of the mountain and the strong ground pressure on the working face is discussed. The sudden increase in the strain peak point of the horizontally distributed optical fiber strain curve can be used to distinguish the strong ground pressure. At the same time, this conclusion is verified by comparing the measured underground ground pressure values. The research results can promote the application of optical fiber sensing technology in the field of mine engineering.

## 1. Introduction

With the depletion of coal resources in the eastern mining areas and the intensification of the contradiction between resources and the environment in central mining areas, China’s coal production capacity is gradually being transferred to the western region [1]. The western mining area has a variety of landforms. Affected by surface gullies, it is a typical gully landform [2,3]. Compared to the general shallow buried coal seam [4], due to the influence of the surface gully, the mining of the working face is generally faced with the problems of mine stress concentration, strong mine pressure manifestation, and support difficulties, which seriously affect the safe and efficient operation of the mine [5].

At present, experts and scholars have carried out some research on the law of the ground pressure behavior of the working face of shallow coal seams through surface valley mining. Wang et al. [6] analyzed the characteristics of the first roof fracture and established a mechanical model of the roof structure during the initial and periodic fracture processes. They revealed the mechanism of sliding and rotational instability of the roof structure. Liu et al. [7] used physical and numerical simulation methods to reveal that the lack of key strata in the process of overlying rock movement during valley terrain is crucial for the mining pressure of fully mechanized mining faces. Yi et al. [8] elucidated the spatiotemporal evolution law of overburden movement caused by shallow, fully mechanized top coal caving and high-strength mining through similar simulation experiments, revealing the movement mode of overburden. This research shows that in the mining process of the working face in a shallow coal seam, the pressure intensity and pressure step are obviously different due to the influence of the surface gully, but the research on the influence of the stability of the mountain structure on the pressure of the working face is less extensive, meaning that it needs further research.

The similar physical model experiment is one of the most important means of studying the movement of overlying strata and the strata behavior of the working face [9,10]. Traditional monitoring [11,12,13] instruments for overburdening deformation include the total station, dial indicator, strain gauge, etc. Most instruments can only monitor the surface deformation of the model, but they cannot monitor the internal deformation of the model. Moreover, the monitoring instruments mostly perform point monitoring, the layout of measuring points is cumbersome, and the sensitivity is low. With the development of optical fiber sensing technology, distributed optical fiber technology based on BOTDA is widely used in similar physical model experiments for the continuous monitoring of overlying rock deformation due to their advantages of high precision, high sensitivity, high reliability, simple layout, and distributed monitoring [14,15,16]. Chai et al. [17] proposed a new method to monitor the deformation of key layers of the overburden and, thus, characterize the pressure in the extraction zone using BOTDA technology. Piao et al. [18] used BOTDA technology to study the strain distribution and motion characteristics of the strata under reamer-pillar mining through similarity simulation experiments, analyzed the stability of the remaining coal pillars in the mining area, and obtained the contour map of the strata strain through calculation. Zhang et al. [19] used a distributed fiber optic sensing technique based on BOTDA to quantitatively access the deformation behavior of overburden rocks from a similar physical model experiment. Many results show that the distributed optical fiber sensing technology based on BOTDA can better characterize the deformation characteristics of overlying rock and the pressure law of the working face.

In this paper, BOTDA technology is applied to the similar physical model experiment, and the DIC technology is combined to monitor the deformation and the motion characteristics of the surface and interior of the model overburden. The motion characteristics of the overburden rock and the law of ground pressure behavior of the working face in a shallow coal seam are comprehensively analyzed when the working face passes through the surface valley. It has referential significance for the mining of working faces under similar conditions.

## 2. Engineering Geology and Production Conditions

Baijigou coal mine is located in the north of the Rujigou mining area in the middle of the Helan mountains. At present, the coal mine mainly mines a 2^−3^ coal seam, with an average thickness of 13.93 m. The mining method is layered mining, and the mining thickness of this layer is 3.0 m. The dip angle of the coal seam is between 4° and 9°, with an average angle of 6°. The relative elevation difference of the surface gullies is about 150.0 m, and the slope of the mountain is between 45° and 75°. The type of coal mining technology is the comprehensive mechanized coal mining technology. The 010204 working face is located in the southern mining area of Baijigou coal mine. The strike length of the working face is 350.0 m, and the dip length is 195.0 m. The characteristics of surface gullies and roadway layout of the 010204 working face are shown in Figure 1.

The immediate roof of the 010204 working face is a gray–black thin-layer siltstone. The thickness of the rock layer is between 1.6 m and 6.6 m, and the average thickness of the rock layer is 4.1 m. The main roof is gray–white thick bedded coarse sandstone with an average thickness of 30.0 m. The direct bottom is siltstone with an average thickness of 0.7 m. The old bottom is dark gray thin bedded siltstone with an average thickness of 23.1 m.

## 3. Similar Physical Model Experiment

### 3.1. Physical Model Design

A similar material model experiment is an effective method for studying the laws of overlying rock failure and working face weighting, which builds a physical model based on the principle of similarity and restores the mining process. The plane model frame with a geometric size of 5.0 × 0.2 × 2.0 m is selected for the similar physical model experiment. Based on the geological conditions of Baijigou coal mine and the size of the experimental model frame, it is determined that when the geometric similarity ratio $C_{l}=l_{p}/l_{m}=200$ (*l_p_* is the prototype size on site, *l_m_* is the model size in the laboratory), the height of the model is appropriate. The average density of the formation is $\gamma_{p}$ = 2.5 g/cm^3^, and the density of similar materials is $\gamma_{m}$ = 1.6 g/cm^3^, while the bulk density similarity ratio is $C_{\gamma }=\gamma_{p}/\gamma_{m}=1.56$. The stress affinity constant is $C_{\sigma }=C_{l}C_{\gamma }=312$, and the time similarity ratio is $C_{t}=\sqrt{C_{l}}=14.14$. When the thickness of the single rock layer of the similar physical model is less than 1.0 cm, it is adverse to the building of the model. Therefore, the model rock layer is appropriately adjusted without affecting the experiment (see Table 1 for the laying layer of the model and the ratio of similar materials).

Based on the actual terrain conditions and geometric similarity ratio, the shape of the model is cut during air-drying. In order to better distinguish the different mining stages in the mining process of the working face, the shape of the surface mountain is simplified and designed as an isosceles trapezoid. The length of the upper line of the isosceles trapezoid mountain is 20.0 cm, the length of the lower line is 236.8 cm, and the height is 108.4 cm. The bottom of the slope on the left side of the mountain is 24.6 cm away from the bottom floor of the model and 27.7 cm away from the left boundary of the model, and the angle of the bottom angle is 45°. The physical model is shown in Figure 2.

### 3.2. Monitoring System Layout

The monitoring system includes the BOTDA optical fiber monitoring system, the DIC digital speckle monitoring system, and the stress monitoring system of the top and bottom of the working face. The monitoring system achieves the monitoring of the changes in stress, as well as strain and displacement on the surface and inside the model.

Before laying similar materials on the model, three distributed optical fibers are embedded inside the model, including two vertical optical fibers and one horizontal optical fiber. The vertical optical fibers of V1 and V2 are 82.0 cm and 210.0 cm away from the left boundary of the model, respectively. The horizontal optical fiber H1 is located in the middle of the subcritical layer of the model, and H1 is 30.0 cm away from the bottom of the model. When the BOTDA monitoring system collects the data, the optical fibers need to form a closed loop, meaning that the three optical fibers are fused with each other. The upper side of the vertical optical fiber V2 is connected to the pump light input end of the optical fiber monitoring equipment, and the right side of the horizontal optical fiber H1 is connected to the detection light output end of the optical fiber monitoring equipment. The red dotted line is the optical fiber buried inside the model, and the red solid line is the external connecting line. The experiment monitoring system deployment is shown in Figure 3.

BOTDA optical fiber sensing technology is based on the principle of Brillouin scattering, using the effect of stimulated Brillouin scattering to obtain the Brillouin frequency shift inside the optical fiber based on the influence of external strain or temperature changes [20]. The Brillouin frequency shift is related to the external strain and ambient temperature, as shown in Equations (1) and (2).
(1)$$\Delta V_{B}=C_{1}\Delta T+C_{2}\Delta \varepsilon$$
(2)$$\Delta \varepsilon =\left[\Delta V_{B}-C_{1}\Delta T\right]/C_{2}$$
where $\Delta V_{B}$ is the Brillouin frequency shift change, $C_{1}$ is the Brillouin temperature coefficient, $C_{2}$ is the strain calibration parameter, ΔT is the difference in temperature change, and Δε is the difference in the strain change.

As the model experiment is carried out indoors, the indoor temperature change is small, and the temperature change range is between −2 °C and 2 °C. When the temperature change is less than 5 °C, the Brillouin frequency shift caused by temperature can be ignored. Therefore, the change in Brillouin frequency shift during the experiment is only related to the change in external strain. The BOTDA sensing technology is shown in Figure 4.

The BOTDA monitoring system used in the experiment is the NBX-6055 Brillouin Time Domain Stress Analyzer produced by Neubrex Company, Kobe, Japan. Its monitoring parameters are set as a 5.0 cm spatial resolution, a 1.0 cm sampling interval, and 2^16^ averaging times. The output probe power is 0 dBm, the output pump power is 30 dBm, and the frequency range is set from 10.60 GHz to 11.00 GHz. The distributed optical fiber used in the experiment is a 2.0 mm diameter single-mode polyurethane tight sleeve optical fiber. The elastic modulus, shear modulus, and density of the optical fiber are 300.0 kPa, 3.3 kPa, and 25.0 g/cm^3^, respectively. The strain calibration parameter *C_2_* of the optical fiber is 0.05 MHz/10^−6^.

### 3.3. Experiment Steps

The width of the boundary coal pillar on both sides of the working face is 8.0 cm, and the mining height of the working face is 3.0 cm. The cut length is 10.0 cm. The working face is excavated from left to right, and the excavation distance is 3.0 cm each time. The working face has been excavated 82 times in total, and the cumulative mining distance is 256.0 cm. During the excavation of the working face, the back ditch mining stage, the peak mining stage, and the trench mining stage are successively passed. The mining distance of back ditch mining is 0.0 cm to 127.0 cm, that of mountaintop mining is 127.0 cm to 148.0 cm, and that of trench mining is 148.0 cm to 256.0 cm. After each excavation of the working face, the experiment personnel first need to wait for the overburden to be fully stable, and then they collect data from each monitoring system and, finally, carry out the next excavation after data collection.

## 4. Experiment Phenomena and Experiment Data

### 4.1. Model Experiment Phenomena

In the back ditch mining stage, the overburden rock was broken for six times. The breaking distances of overburden are 31.0 cm, 21.0 cm, 12.0 cm, 18.0 cm, 9.0 cm, and 33.0 cm, respectively. Every time the overburden is broken, irregular trapezoidal rock blocks will be generated. The “trapezoidal” rock blocks are numbered from low to high as ① to ⑤, as shown in Figure 5. When the working face is advanced to 31.0 cm, the main roof is broken for the first time, and it is a “voussoir beam” hinged structure. Mining cracks develop to the surface, and the working face is pressurized for the first time. When the working face is advanced to 52.0 cm, the subcritical layer is broken for the first time, and the working face is pressurized for the first time. The irregular trapezoidal rock block of number ① was generated and overturned to the left. When the working face is advanced to 64.0 cm, the irregular trapezoidal rock block of number ② is generated and occludes with the low rock block of number ①, and the mining cracks are gradually closed. Then, as the working face continues to advance, the main roof periodically breaks and collapses into a “voussoir beam” structure, and the subcritical layer above the working face is a “cantilever beam” structure. Irregular trapezoidal rock blocks are continuously generated to a high position along with the periodic fracture of subcritical layers, causing the instability of the lower “voussoir beam” structure and caving into the goaf.

In the mountaintop mining stage, when the working face is advanced to 142.0 cm, the working face is pressurized for the ninth cycle. Tension cracks appear in front of the working face, and the breaking angle is 59°. The tension crack develops to 11.0 cm in front of the working face, and the crack height is 60.0 cm. The mountain body deflects and sinks to the left and shows “imbalance”. The mining cracks produced in the back ditch mining stage are gradually closed. During the process of mountain deflection, the cracks did not develop to the main key stratum, the main key stratum did not obviously break, and the main key stratum and its overlying strata deflected in the same direction as the mountain, as shown in Figure 6.

In the trench mining stage, with the advance of the working face, the overlying rock collapses layer by layer from bottom to top. The mountain gradually turns to the right, and the advance tension crack gradually closes. The communication between the cracks developed from the bottom to the top, and the pre-tensioning cracks led to the large-scale fracture and collapse of the overlying rock. The sliding phenomenon occurred at the slope bottom in the trench mining stage, as shown in Figure 7.

During the mining period of the working face, a total of 14 cycles of weighting occurred. Among them, there are eight instances of cyclic weighting in the back ditch mining stage, and the average cyclic weighting step is 11.6 cm. There is periodic weighting at the mountaintop mining stage, and the periodic weighting step is 18.0 cm. There are five instances of cyclic weighting in the trench mining stage, and the average interval of cyclic weighting is 19.2 cm (see Table 2 for the pressure on the working face).

### 4.2. Experiment Data

(1)Horizontal optical fiber H1

The strain experiment results of horizontal optical fiber H1 are shown in Figure 8. When the working face is advanced to 52.0 cm, the strain curve of the optical fiber significantly changes. The strain curve shows a single peak shape, and the peak value of strain is 4191 µε. Then, with the advancement of the working face, the peak strain increases. When the working face is advanced to 82.0 cm, the strain curve changes from a single peak to a double peak. The left and right peaks of the strain curve are 4310 µε and 12,700 µε, respectively. When the working face is advanced from 82.0 cm to 139.0 cm, the size and location of the left peak of the strain curve are basically unchanged, and the peak value is 3382 µε. The right peak value migrates to the right with the mining of the working face and gradually decreases. When the working face is advanced to 139.0 cm, the peak value on the right side of the strain curve decreases to a minimum of 4084 µε, and the peak shifts to the left. When the working face is advanced to 142.0 cm, two new peaks are added to the strain curve. The newly increased peaks from left to right are 9593 µε and 2100 µε, respectively. When the working face is advanced to 238.0 cm, the peak of the strain curve suddenly increases again, and the strain suddenly increases to 14,987 µε.

(2)Vertical fiber optic

The strain experiment results of vertical optical fiber V1 are shown in Figure 9, and the gray part is the relative position of 2^−3^ coal seam. When the working face is mined for a certain distance, the negative peak strain appears in the lower part of the optical fiber as the working face gradually approaches the optical fiber V1. When the working face advances to 73.0 cm, the peak value of negative strain reaches −481 µε, and the influence range extends from 0.0 cm to 28.9 cm, as shown in Figure 9a. When the working face passes through the optical fiber V1, the strain curve of the optical fiber changes from negative peak strain to positive peak strain, and it then increases periodically. When the working face is advanced to 91.0 cm, the maximum positive peak strain is 28,700 µε, as shown in Figure 9b. When the working face is far away from the optical fiber V1, the peak strain gradually decreases and tends to be stable until it decreases to 11,104 µε, as shown in Figure 9c.

The strain experiment results of vertical optical fiber V2 are shown in Figure 10. When the working face is gradually close to V2 but far away from V1, the positive peak strain appears in the strain curve. When the working face is advanced to 142.0 cm, two obvious peaks appear in the strain curve. The two peaks from bottom to top are 2361 µε and 2961 µε, respectively. Then, with the continuous advance of the working face, the positive strain bimodal of the strain curve gradually decreases. The strain curve moves to the negative axis of the coordinate axis. The vertical optical fiber V2 is pulled in the middle and pressed up and down, as shown in Figure 10a. When the working face passes through the vertical optical fiber V2, the positive peak strain of the strain curve continuously increases. When the working face is advanced to 238.0 cm, the maximum peak strain reaches 10,533 µε, as shown in Figure 10b. When the working face is far away from the vertical optical fiber V2, the peak strain gradually decreases and tends to be stable, with a stable peak strain value of 3178 µε, as shown in Figure 10c.

## 5. Analysis of Results

### 5.1. Fibre Optic Characterization of Critical Layer Breakage Patterns

(1)Back ditch mining stage

As shown in Figure 11, when the working face is advanced to 52.0 cm, the subcritical layer is broken for the first time. Mining fissures developed to the surface, forming an irregular trapezoidal rock block number ①. Due to the lack of horizontal binding force on the free side of the slope, rock block of number ① overturned to the left and sank. The strain curve of the horizontal fiber H1 produces a positive No. 1 peak strain, which reaches 4191 µε. When the working face is advanced to 82.0 cm, the subcritical layer is broken again. The high irregular trapezoidal rock block of number 3 has generated and overturned to the left, and the rock block of number 3 and the low rock block of number ② occluded each other. To a certain extent, it prevented the overturning of rock blocks of numbers ① and ②, as well as the synchronous sinking and compaction of rock blocks of numbers ① and ②. The strain curve of horizontal optical fiber H1 changes from a single peak to a double peak under the influence of the rock block of number ③. The No. 1 peak on the left side of the strain curve slightly increased to 4231 µε, and the No. 2 peak on the right side reached 12,700 µε. The peak value 2 is located 4.7 cm behind the working face, corresponding to the position of the longitudinal crack of the rock block of number ③. The sudden increase in the peak of the strain curve is a manifestation of the breaking of the subcritical layer.

As shown in Figure 12, the subcritical layer breaks periodically as the working face advances. The mining-induced fractures are continuously developed to a high position, and irregular trapezoidal rock blocks are generated periodically. When the high-level “trapezoid” rock block is generated, the low-level fracture will gradually close, and the overturning degree of the low-level rock block will be weakened and gradually stabilized. The extent and location of the No. 1 peak strain are basically unchanged. The position of the No. 2 peak strain moves forward to the right with the advance of the working face, and the peak value gradually decreases and tends to be stable. While the working face was advancing from 82.0 cm to 124.0 cm, the No. 2 peak strain continuously decreased and remained stable after decreasing from 9294 µε to 5693 µε. The strain curve of the optical fiber in the back ditch mining stage shows that the horizontal optical fiber can monitor the breaking position of the subcritical layer. The peak strain will migrate to the right with the periodic breaking of the subcritical layer. The position of peak strain reflects the breaking position of the subcritical layer.

(2)Mountaintop mining stage

As shown in Figure 13a–c, during the working face advance from 127.0 cm to 139.0 cm, the position of the No. 2 peak of the optical fiber shifts to the left, and the peak strain gradually decreases. The peak strain decreased from 5694 µε to 4083 µε. The strain curve has a wavelet peak at 53.3 cm in front of the working face, and the peak strain is 356 µε. At this time, the mining fissures in the upper part of rock block number ⑤ are gradually closed, and the mountain has a tendency to deflect to the left. The cracks at the bottom of the slope mined toward the trench are generated. Compared to the strain nephogram of DIC, the strain at the mining fracture of back ditch mining gradually decreases. The local strain increases at the bottom of the trench mining slope. The strain of DIC has the same trend as the strain curve of the optical fiber. The correctness of optical fiber sensing is further verified.

As shown in Figure 13d–f, when the working face advances to 142.0 cm, the strain curve of the optical fiber adds two new peaks on the original basis. The No. 3 peak and No. 4 peak are 25.4 cm and 56.2 cm ahead of the working face, and the corresponding peak strains are 9593 µε and 2140 µε, respectively. The advanced tension crack is generated, and the crack develops on the surface of the trench mining slope. The working face is pressurized for the 8th cycle. The mountain deflected to the left and sank, resulting in the “imbalance” of the mountain. Compared to the strain nephogram of DIC, the strain decreases at the mining fracture of back ditch mining and increases at the advanced tension fracture. The above analysis shows that the horizontal optical fiber can monitor the migration trend of the mountain. The shift and decrease in the peak of the optical fiber strain curve is the precursor of the “imbalance” of the mountain.

(3)Trench mining stage

As shown in Figure 14, while the working face was advancing from 151.0 cm to 235.0 cm, the mountain began to turn and sink to the right. The tension crack gradually closes, and the overlying rock bends, breaks, and collapses from bottom to top. The overall tension degree of the horizontal optical fiber H1 gradually approaches the same trend, and the strain curve of the optical fiber presents a “saddle” shape. When the working face is pushed to 238.0 cm, the subcritical layer is broken, and the working face is pressurized for the 14th cycle. The longitudinal crack development is connected to the advance tension crack, and the sliding phenomenon occurs at the slope bottom of the trench mining. The horizontal optical fiber H1 is affected by the tensile stress at point D, and there is an obvious No. 1 peak. The peak of the fiber strain curve is 14,897 µε. The peak strain of the fiber increases by 159.2% when the working face advances to 238.0 cm compared to when it advances to 235.0 cm. The strain change corresponding to DIC also has the same phenomenon, and there is a sudden increase in strain at the slope slip.

### 5.2. Vertical Fiber Optic Frequency Shift Change to Pressure Characterization

When the working face is mined, the original rock stress of the overlying rock is redistributed. According to the definition of optic fiber frequency shift variation in the document [21], the size of the optic fiber frequency shift variation is used to characterize whether or not incoming pressure is occurring at the working face. The definition of fiber frequency shift variation is
(3)$$D_{j}=\frac{1}{n}\left(\sum_{i=1}^{n}|{\mathrm{BFS}}_{j}|-\sum_{i=1}^{n}|{\mathrm{BFS}}_{j-1}|\right)$$
where *n* is the total number of sampling points for effective monitoring distance of the optical fiber, *j* is the number of working face excavations, $\left|{\mathrm{BFS}}_{j}\right|$ is the absolute value of the frequency shift at a sampling point of the optical fiber, and $D_{j}$ is the variation degree of the optical fiber frequency shift of the jth excavation of the working face.

When the fiber frequency shift variation is more than 20 MHz, it is considered that the first pressure or cycle pressure occurs on the roof of the working face. When the fiber frequency shift variation is more than 60 MHz, it is considered that the roof pressure of the working face is intense.

The contrast information between the variation in the optical fiber frequency shift of vertical optical fiber and the working resistance of the support on the roof of the working face is shown in Figure 15. In the mining process of the working face, there are 15 peaks of optical fiber frequency shift, which are 31.0 cm, 52.0 cm, 64.0 cm, 73.0 cm, 82.0 cm, 91.0 cm, 103 cm, 112.0 cm, 124.0 cm, 142.0 cm, 160.0 cm, 193.0 cm, 211.0 cm, 220.0 cm, and 238.0 cm, respectively. The characterization of the pressure on the optical fiber is consistent with the pressure on the working face in the experiment, which further shows that the frequency shift variation in the optical fiber can better reflect the pressure on the working face.

The experiment shows that the initial pressure step of the working face is 31.0 cm. The average periodic weighting step in the back ditch mining stage is 11.6 cm. The average periodic weighting step in the mountaintop mining stage is 18.0 cm. The average periodic weighting step in the trench mining stage is 19.2 cm. The average periodic weighting steps of mountaintop mining and trench mining increased by 55.2% and 65.5%, respectively, compared to the average periodic weighting step of back ditch mining. The periodic weighting step of the working face is characterized by “small periodic weighting step of back ditch mining and large periodic weighting step of other stages”, as shown in Figure 16.

According to the experiment phenomenon and the pressure situation of the working face, the mine pressure behavior law of the working face passing through the surface valley is obvious in stages. The weighting characteristics are as follows: the weighting is frequent in the back ditch mining stage, and the dynamic mine pressure is obvious. The mine pressure behavior in the mountaintop mining stage and the trench mining stage is relatively mild, but the mine pressure behavior in the working face is strong when mountain deflection and slope slip occur. The phased characteristics of the strata behavior law of the working face show that the stability of the mountain structure has a significant impact on the periodic weighting of the working face.

### 5.3. Horizontal Optical Fiber Characterization of Mountain Stability

We extract the peak point of the strain curve of the horizontal optical fiber H1 in Figure 11, Figure 12, Figure 13 and Figure 14 and draw the change curve of the peak point of the horizontal optical fiber H1 during the mining process of the working face, combined with the DIC strain nephogram for joint analysis. As shown in Figure 17, panel A shows the overburden fracture in the back ditch mining stage. Panel B shows the deflection of the mountain during the mountaintop mining stage. Panel C shows the slope sliding during the trench mining stage.

During the back ditch mining stage, the stability of the mountain structure is good at the initial stage of the working face excavation, and the peak point of strain hardly changes. When the overburden is obviously broken, the peak point of strain of the optical fiber suddenly increases to peak point 1. The strain at the peak point increases from 4304 µε to 17,336 µε, and the growth rate of strain at the peak point is 302.7%. With the advancement of the working face, the strain at the peak point gradually decreases. This process occurs with the periodic fracture of the overlying rock. When the mountain deflects, the peak point of strain of the optical fiber suddenly increases again to peak point 2. The strain at the peak point increases from 4084 µε to 9584 µε, and the growth rate of the strain at the peak point is 134.7%. Then, it enters the trench mining stage, and the strain at the peak point continues to decrease. When the slope slip occurs, the peak point of strain suddenly increases to peak point 3. The strain at the peak point increases from 5980 µε to 14,944 µε, and the growth rate of strain at the peak point is 149.9%. It can be seen that the strain at the peak point of the horizontal optical fiber H1 will increase significantly when the mountain is broken, the mountain deflects, and the slope slides.

In combination with the above peak point changes, the changes in the peak point of the horizontal optical fiber H1 are compared to the pressure on the working face, as shown in Figure 18. The peak point of the horizontal optical fiber H1 increases significantly when the overlying rock breaks, the mountain deflects, and the slope slides. The corresponding working face has strong ground pressure behavior. The ground pressure intensities of overburden failure, mountain deflection, and slope sliding are 88.4 MHz, 97.6 MHz, and 69.9 MHz, respectively, and the average periodic weighting intensity is 85.3 MHz. Compared to the average periodic weighting strength during other weighting periods, the average increase rate is 99.9%. The peak point of the sudden increase corresponds to the strong ground pressure behavior position of the working face (as shown in the gray dotted box in the figure).

The comparative analysis results show that the strong ground pressure behavior of the working face can be characterized by the horizontal optical fiber. When the peak point of the horizontal optical fiber has an obvious sudden increase in strain and the growth rate of strain is greater than 100%, the working face will exhibit strong ground pressure behavior.

### 5.4. Field Measurement of Mine Pressure in the Working Face

The 010204 working face started mining on 3 November 2022. The pressure data of 14 hydraulic supports with support numbers of 2, 10, 20, …, 110, 120, and 129 are selected to analyze the movement and failure of the roof and the stress of the supports. The rated resistance and initial support force of the supports at the working face are 36.5 MPa and 26.5 MPa, respectively. When the resistance of the supports reaches 90% (33.0 MPa) of the rated working resistance, that is, the average coefficient of load increase exceeds 1.25, the supports will give an alarm (see Table 3 for the average coefficient of load increase and alarm rate at different mining stages).

The average coefficient of load increase in the support in the back ditch mining stage is 1.37, and the alarm rate is 49.0%. The support resistance is significantly higher than that in other mining stages, and the support alarm is frequent. It shows that the periodic weighting in the back ditch mining stage is more severe than that in other mining stages. The average coefficient of load increase in the supports in the mountaintop mining stage and the trench mining stage are both 1.22, making them less than the alarm value of 1.25. The alarm rate is also low, and the mine pressure behavior is relatively mild.

The measured pressure characteristics of the field are basically consistent with the model experiment pressure characteristics, which shows that the distributed optical fiber is correct and feasible to characterize the pressure. It provides a new method for mine pressure monitoring.

## 6. Conclusions

(1)The horizontal optical fiber can monitor the breaking position of the subcritical layer and the stability of the mountain structure. The peak of the strain curve increases after the fracture of the subcritical layer, and the position of the peak reflects the fracture position of the subcritical layer. The shift and decrease in the peak strain is the precursor to the “imbalance” of the mountain.(2)The periodic weighting step of the working face is characterized by “small periodic weighting step of back ditch mining, and large periodic weighting step of mountaintop mining and trench mining”. The average periodic weighting step distances of mountaintop mining and trench mining are 55.2% and 65.5% higher than that of back trench mining, respectively.(3)The strata behavior law of the working face passing through the surface gullies is obvious in stages. The weighting performance is that the periodic weighting of back ditch mining is frequent, and the dynamic load mine pressure is serious. The periodic weighting in the mountaintop mining and trench mining stages is relatively mild. The stability of the mountain structure has a significant impact on the pressure on the working face. When the mountain deflects or the slope slides, the ground pressure behavior of the working face is strong.(4)A new method for characterizing the strong ground pressure behavior of the working face is proposed. The strong ground pressure of the working face is characterized by the strain change in the horizontal optical fiber. When the peak point of strain of the horizontal optical fiber undergoes a significant sudden increase and the growth rate of the strain peak point exceeds 100%, the working face will have strong ground pressure behavior.

## Figures and Tables

**Figure 1 sensors-23-09063-f001:**
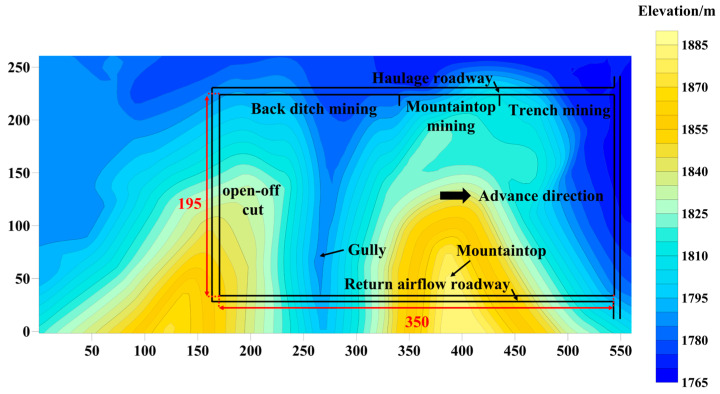
Characteristics of surface gullies and the roadway layout in 010204 working face.

**Figure 2 sensors-23-09063-f002:**
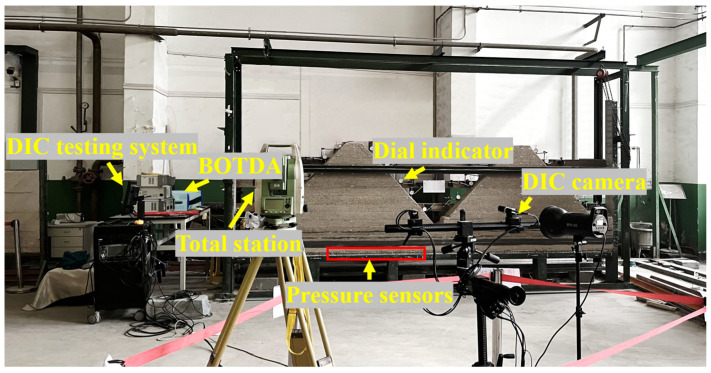
Similar physical model.

**Figure 3 sensors-23-09063-f003:**
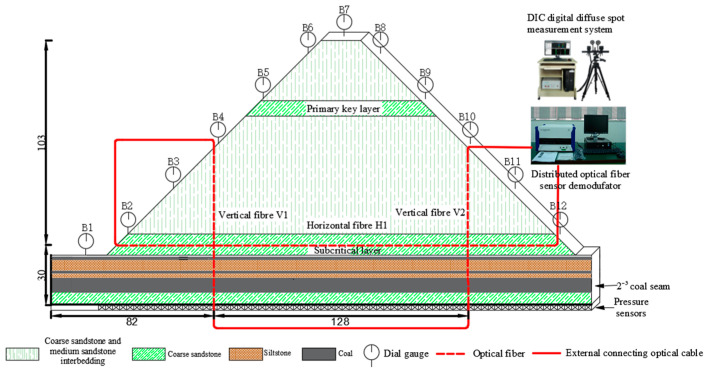
Experiment monitoring system.

**Figure 4 sensors-23-09063-f004:**
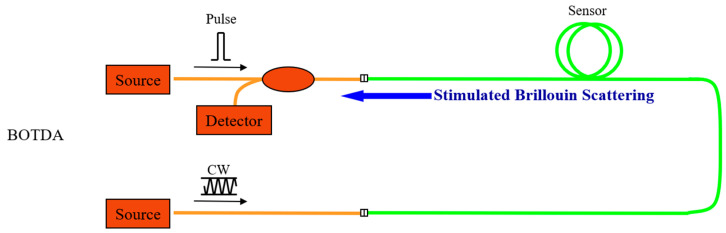
BOTDA sensing technology.

**Figure 5 sensors-23-09063-f005:**
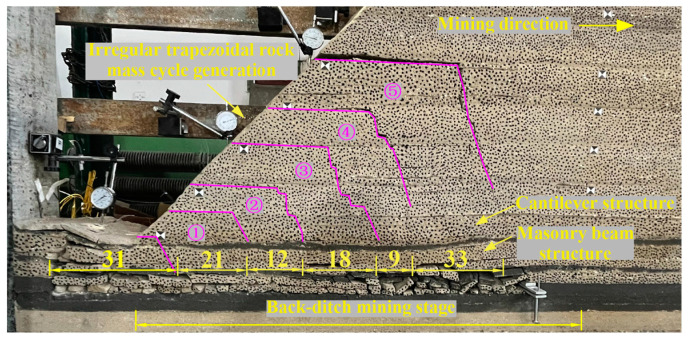
Caving characteristics of overlying strata in back ditch mining stage.

**Figure 6 sensors-23-09063-f006:**
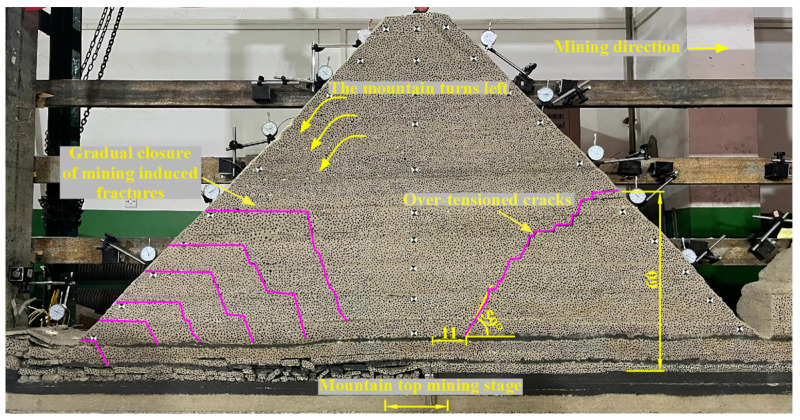
Caving characteristics of overlying strata in mountaintop mining.

**Figure 7 sensors-23-09063-f007:**
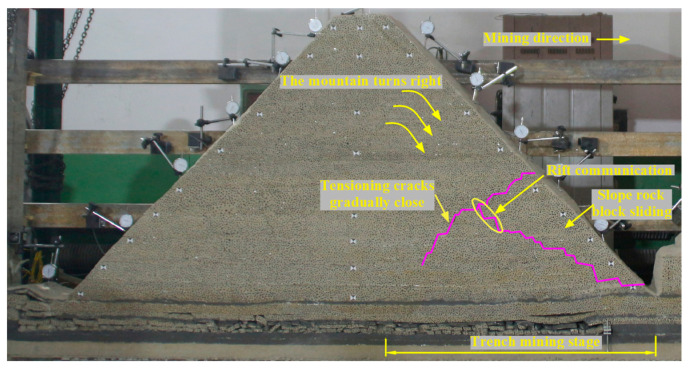
Caving characteristics of overlying strata in the trench mining stage.

**Figure 8 sensors-23-09063-f008:**
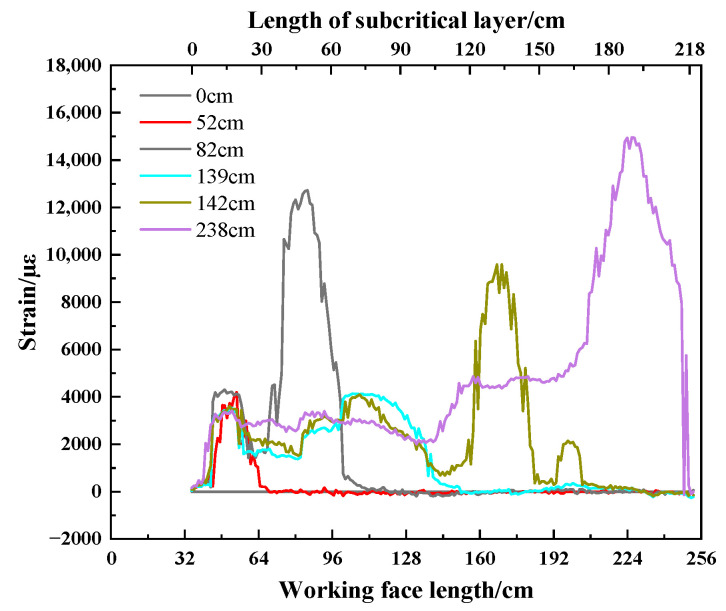
Strain curve of the horizontal optical fiber H1.

**Figure 9 sensors-23-09063-f009:**
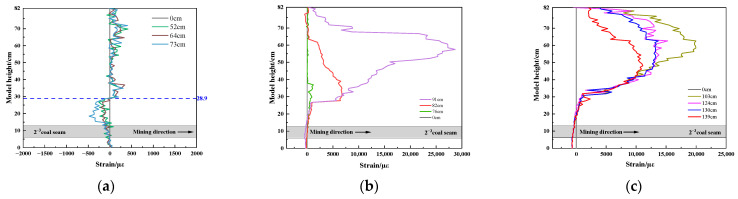
Strain curve of vertical optical fiber V1: (**a**) working face close to V1; (**b**) working face passing through V1; (**c**) working face away from V1.

**Figure 10 sensors-23-09063-f010:**
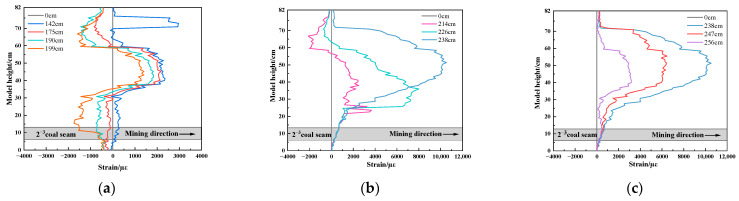
Strain curve of vertical optical fiber V2: (**a**) working face close to V2; (**b**) working face passing through V2; (**c**) working face away from V2.

**Figure 11 sensors-23-09063-f011:**
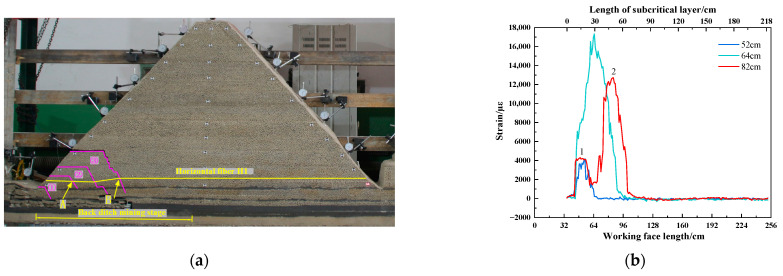
Working face advancing from 52.0 cm to 82.0 cm: (**a**) diagram of subcritical layer breakage; (**b**) fiber optic strain curve variation diagram.

**Figure 12 sensors-23-09063-f012:**
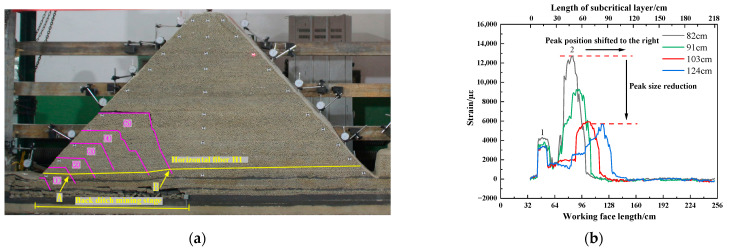
Working face advancing from 82.0 cm to 124.0 cm: (**a**) diagram of sub critical layer breakage; (**b**) fiber optic strain curve variation diagram.

**Figure 13 sensors-23-09063-f013:**
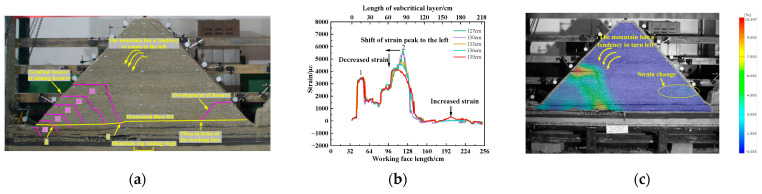
Mountaintop mining stage: (**a**) breakage of sub critical layers before mountain deflection; (**b**) change in strain curve before mountain deflection; (**c**) surface strain changes before mountain deflection; (**d**) breaking of subcritical layers after mountain deflection; (**e**) changes in the fiber optic strain curve after mountain deflection; (**f**) surface strain changes after mountain rotation.

**Figure 14 sensors-23-09063-f014:**
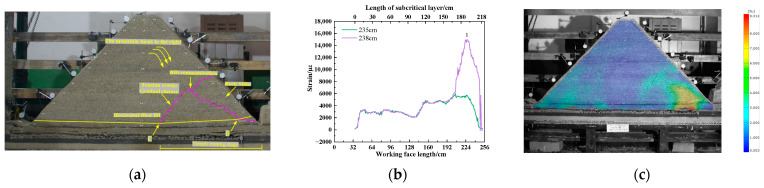
Stage of trench mining: (**a**) advance to 238.0 cm overlying rock fracture situation; (**b**) fiber optic strain change; (**c**) surface strain variation in overlying rock.

**Figure 15 sensors-23-09063-f015:**
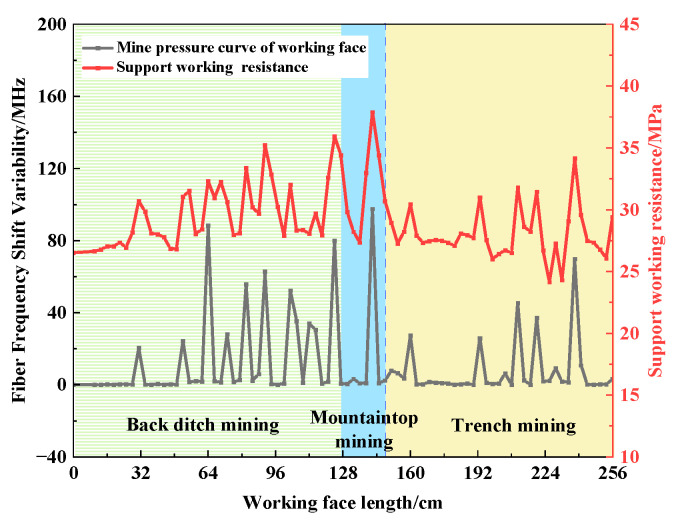
Comparison between fiber frequency shift variation and support resistance.

**Figure 16 sensors-23-09063-f016:**
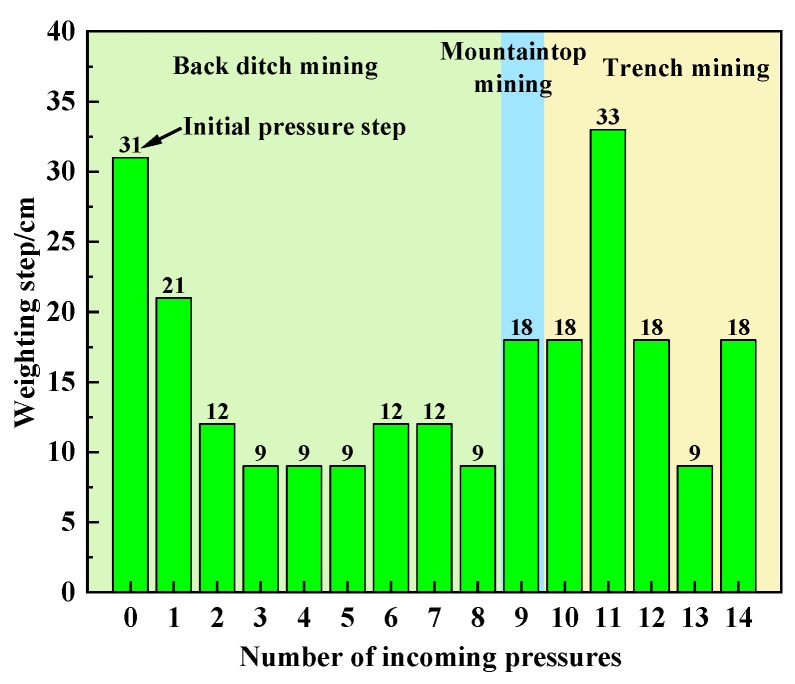
Periodic weighting step of working face.

**Figure 17 sensors-23-09063-f017:**
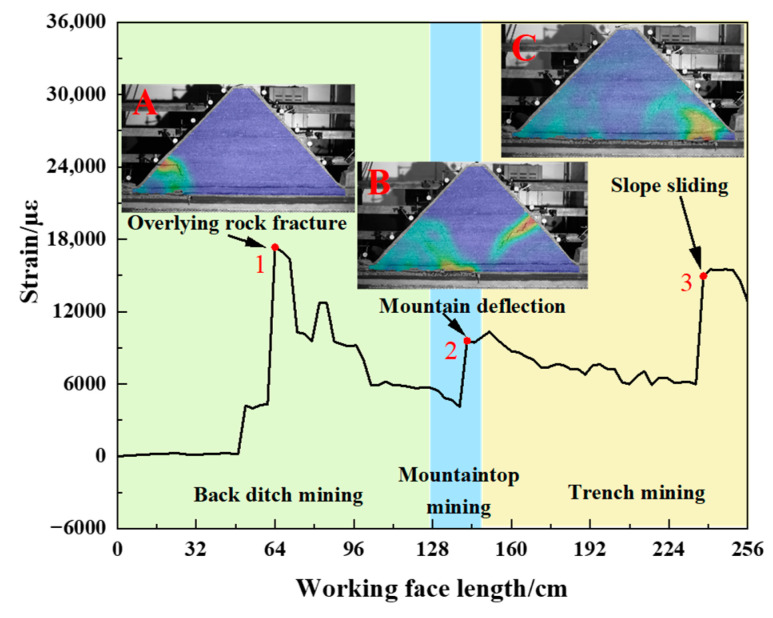
Variation curve of peak point of strain of horizontal optical fiber H1.

**Figure 18 sensors-23-09063-f018:**
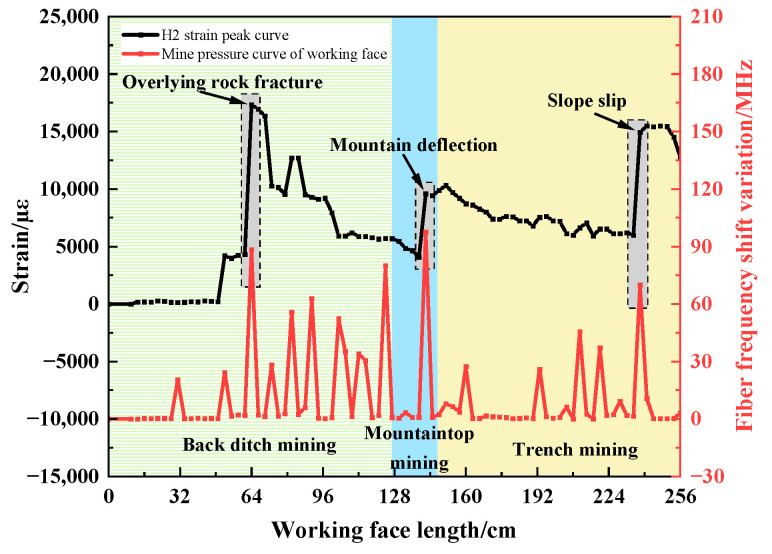
Comparative analysis diagram of peak point change in horizontal optical fiber and pressure on working face.

**Table 1 sensors-23-09063-t001:** Proportioning of similar materials.

Lithology of Simulated Stratum	Model Thickness (cm)	Cumulative Thickness (cm)	Matching Ratio (Sand:Gypsum:Mica Powder)
Coarse sandstone and medium sandstone interbedding	97.5	133.0	7:2:8 and 8:2:8
Coarse sandstone	10.8	59.4	7:2:8
Fine sandstone	1.0	24.5	8:3:7
2^−1^ coal seam	1.3	23.5	
Siltstone	1.6	22.2	7:2:8
Coarse sandstone	4.1	20.7	7:2:8
2^−2^ coal seam	1.1	16.6	
Siltstone	1.3	15.5	7:2:8
Coarse sandstone	1.3	14.3	7:2:8
2^−3^ coal seam	7.0	13.0	
Coarse sandstone	6.0	6.0	7:2:8

**Table 2 sensors-23-09063-t002:** Working face pressure situation.

Mining Stage	Number of Pressure Applications	Advance Distance (cm)	Weighting Step (cm)
Back ditch mining stage	Initial pressure	31.0	31.0
	1st cycle pressure	52.0	21.0
	2nd cycle pressure	64.0	12.0
	3rd cycle pressure	73.0	9.0
	4th cycle pressure	82.0	9.0
	5th cycle pressure	91.0	9.0
	6th cycle pressure	103.0	12.0
	7th cycle pressure	115.0	12.0
	8th cycle pressure	124.0	9.0
Mountaintop mining stage	9th cycle pressure	142.0	18.0
Trench mining stage	10th cycle pressure	160.0	18.0
	11th cycle pressure	193.0	33.0
	12th cycle pressure	211.0	18.0
	13th cycle pressure	220.0	9.0
	14th cycle pressure	238.0	18.0

**Table 3 sensors-23-09063-t003:** Average coefficient of load increase and alarm rate of support at each stage.

Mining Stage	Average Coefficient of Load Increase	Bracket Average Alarm Rate
Back ditch mining stage	1.37	49.0%
Mountaintop mining stage	1.22	21.7%
Trench mining stage	1.22	32.1%

## Data Availability

All the data, models, or codes that support the findings of this study are available from the corresponding author upon reasonable request.

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
