# Peer review of "Research on Similarity Simulation Experiment of Mine Pressure Appearance in Surface Gully Working Face Based on BOTDA"

_sensors, 2023, doi:10.3390/s23229063_

Round 1
Reviewer 1 Report
Comments and Suggestions for Authors
The paper presented by the authors is rather interesting and timely, containing important practical knowledge. However, I would highlight a number of major and minor issues that need to be addressed before publication:
1. The introduction mentions distributed fiber optic sensors in general. It is not stated why this particular type of fiber sensor is selected. Since the experiment takes place in the laboratory on a reduced copy of a real object, it is necessary to have a sufficiently high spatial resolution. The BOTDA/BOTDR technology, like all time domain approaches, can provide fairly low spatial resolution. In this regard, the question arises: why an optical reflectometry in the frequency domain is not used? The OFDR method has high resolution and has achieved large dynamic ranges due to the signal processing and amplification [https://doi.org/10.3390/s18041072, http://dx.doi.org/10.1134/S0020441223050172].
2. The sensory experimental setup is not described. If this is a commercial system, its brand and specification must be given.
3. The probing regimes are not described - what pulses are sent into the fiber line, how the signal is processed - are Lorentzian Curve Fitting (LCF) and/or Backward Correlation method (BWC) used?
4. After reading the work, I did not have a clear understanding of whether the problem can be scaled to the size of a laboratory without unacceptable simplifications and assumptions. Please write more about this.
5. Does the graph in the figure 17 give the correct values? Individual peaks resemble an algorithmic error of Lorentzian curve fitting.
6. Some pictures are located right at the end of the section, I suggest moving them higher, right after the first mention.
Author Response
Response 1: The BOTDA distributed fiber optic sensing technology, with its high resolution (spatial resolution can reach 5.0cm), can continuously monitor the internal deformation of the overlying rock, achieve monitoring of overlying rock deformation and characterization of working face pressure. Therefore, this manuscript selects distributed fiber optic sensing technology based on BOTDA to monitor the strain inside similar physical models. I have made revisions to lines 58 to 72 on page 2 of the manuscript, explaining the advantages of BOTDA technology and its widespread application in monitoring overlying rock deformation.
Response 2: The BOTDA monitoring system used in the experiment is the NBX-6055 Brillouin Time Domain Stress Analyzer and computer produced by Neubrex Company in Japan. Its monitoring parameters are set as 5.0 cm spatial resolution, 1.0 cm sampling interval, and 216 averaging times. I have added the BOTDA manufacturer and specifications on lines 162 to 163 of page 5 of the manuscript.
Response 3: The detection mechanism of BOTDA technology in this manuscript is Pulse Pre Pump Brillouin Time Domain Analysis (PPP BOTDA). By loading appropriate pulse pre-pump light before introducing pulse light (pump light), phonons are pre-excited. And by appropriately setting the power ratio of the pulse light to the pulse pre-pump light, excess output power can be reduced. Thus achieving high spatial resolution and high-precision monitoring. The BOTDA sensing technology diagram can be found on page 5, line 160 of the manuscript. I have added the pulse signal parameters sent into the fiber optic line on lines 165 to 166 of the 5 page of the manuscript. The BOTDA monitoring system uses Lorentz curve fitting. The system uses software to fit the collected signal, then outputs the corresponding data file, and finally calculates the strain change manually through formula â–³VB=C1â–³T+C2â–³ε .
Response 4: Similar material model experiment is an effective method for studying the laws of overlying rock failure and working face weighting, which builds a physical model based on the principle of similarity and restores the mining process. The relative height difference between the surface valleys on site is about 150m, and the geometric size of the model frame is 5.0×0.2×2.0m. Therefore, it is determined that when the geometric similarity ratio is Cl=lp/lm=200 ( lp is the prototype size on site, lm is the model size in the laboratory), the total height of the physical model is more suitable. The average density of the formation is γp=2.5g/cm3, and the density of similar materials is γm= 1.6g/cm3, the bulk density similarity ratio is Cy=γp/γm=1.56. The stress affinity con-stant is Cσ=Cl Cy =312, and the time similarity ratio is Ct= Cl½. I have provided detailed explanations on lines 100 to 112 on page 3 of the manuscript.
Response 5:The graph in Figure 17 shows the correct values. The graph curve is drawn by extracting the data of the peak strain point of the horizontal optical fiber H1 during each excavation of the working face.
Response 6: I have moved some of the images to the first mentioned location, such as Figure 4, Figure 17, and Figure 18 in the manuscript, to line 160 on page 5, line 426 on page 14, and line 454 on page 15, respectively.

Reviewer 2 Report
Comments and Suggestions for Authors
In this paper, the authors conducted similarity simulation experiment of mine pressure appearance in surface gully working face based on BOTDA, which promote the application of optical fiber sensing technology in the field of mine engineering. The following are the questions in this manuscript:
(1) In Figure 3, it is suggested that the red dotted and solid lines correspond to the optical fibers and the external connecting lines, respectively.
(2) In Figure 4, there are two figures, the titles of the figure should be given.
(3) In Figure 9(a), the “73cm” does not match the above and should be changed to “72cm”.
(4) In page 10 line 248, the “close to V2 but far away from V2” does not match the actual situation and should be changed to “close to V2 but far away from V1”.
(5) In page 11 line 258, the author only mentioned that “the peak strain gradually decreases and tends to be stable”. It is suggested to add the stable value after the peak strain decreases.
(6) In Figure 11(b), the author provides the fiber optic strain curves when the working face advanced to 52.0cm and 82.0cm. It is suggested to add the curve when the working face advanced to 64.0cm.
(7) In page 13 line 347, the author mentioned that “the rate of increase in peak strain is 159.2%”, lacking the premise that the working face advanced from 235cm to 238cm.
(8) In page 16, “Figure 17” should be changed to “Figure 18”.
(9) There are some issues with the layout of this article, such as the small sizes of the six images in Figure 13 and the three images in Figure 14 that cannot be seen clearly.
(10) There are many formatting problems in the article, such as the use of citation symbols in papers in line 31 and 33.
Author Response
Response 1: I have made modifications to Figure 3 on line 142 on page 5 of the manuscript. The red dashed lines and red solid lines in Figure 3 represent the optical fibers inside the model and the connecting optical lines outside the model, respectively.
Response 2: I have replaced Figure 4 to better reflect the detection mechanism of BOTDA sensing technology. Figure 4 is located on page 5 of the manuscript, line 160.
Response 3: Thank you. The advance distance data in the manuscript is incorrect. I have revised “When the working face advances to 72.0cm” in line 254 on page 9 of the manuscript to “When the working face advances to 73.0cm”.
Response 4: I have changed from “When the working face is gradually closed to V2 but far away from V2” to “When the working face is gradually closed to V2 but far away from V1” in line 266 on page 9 of the manuscript.
Response 5: In response to this issue, I have revised from “the peak strain gradually decreases and tends to be stable” to “the peak strain gradually decreases and tends to be stable,with a stable peak strain value of 3178 uε” in lines 276 to 277 on page 9 of the manuscript.
Response 6: I have added the strain curve when the working face advances to 64.0cm in Figure 11 (b), located on page 10, line 302 of the manuscript.
Response 7: I added the prerequisite for a peak strain growth rate of 159.2% for the strain curve when the working face advances to 238.0cm, which is “The peak strain of the fiber increases by 159.2% when the working face advances to 238.0 cm compared to when it advances to 235.0 cm”. located on page 12, lines 365 to 366 of the manuscript.
Response 8: I have changed the title of the last image from Figure 17 to Figure 18, located on page 15, line 455 of the manuscript.
Response 9: I have replaced and enlarged the six images of Figure 13 and the three images of Figure 14 to make them clearer, located on page 11, line 347, page 12, line 349, and line 369 of the manuscript, respectively.
Response 10: I have carefully checked the formatting of the manuscript and made modifications to address these issues. For example, the font size of the image title, the coordinate names in the image, and the replacement of professional terms in the manuscript.

Round 2
Reviewer 1 Report
Comments and Suggestions for Authors
The authors correctly addressed my remarks. I think that this paper could be published in present form now.